# Combined Fully Contactless Finger and Hand Vein Capturing Device with a Corresponding Dataset

**DOI:** 10.3390/s19225014

**Published:** 2019-11-17

**Authors:** Christof Kauba, Bernhard Prommegger, Andreas Uhl

**Affiliations:** Department of Computer Sciences, University of Salzburg, Jakob-Haringer-Str. 2, 5020 Salzburg, Austria; bprommeg@cs.sbg.ac.at (B.P.); uhl@cs.sbg.ac.at (A.U.)

**Keywords:** finger vein recognition, hand vein recognition, contactless acquisition device, public vascular pattern dataset, biometric recognition performance evaluation

## Abstract

Vascular pattern based biometric recognition is gaining more and more attention, with a trend towards contactless acquisition. An important requirement for conducting research in vascular pattern recognition are available datasets. These datasets can be established using a suitable biometric capturing device. A sophisticated capturing device design is important for good image quality and, furthermore, at a decent recognition rate. We propose a novel contactless capturing device design, including technical details of its individual parts. Our capturing device is suitable for finger and hand vein image acquisition and is able to acquire palmar finger vein images using light transmission as well as palmar hand vein images using reflected light. An experimental evaluation using several well-established vein recognition schemes on a dataset acquired with the proposed capturing device confirms its good image quality and competitive recognition performance. This challenging dataset, which is one of the first publicly available contactless finger and hand vein datasets, is published as well.

## 1. Introduction

Biometric authentication is gaining more and more attention and replaces traditional authentication methods like passwords, signatures and tokens. It offers higher security and increased user convenience compared to traditional methods. Biometric authentication techniques are based on so-called biometric traits, which are behavioural or physiological characteristics of a person. These biometric traits are unique to every person. The most commonly used biometric traits include fingerprints, face and iris. Recently, vascular pattern based biometrics, especially hand and finger based vascular patterns (usually denoted as hand and finger vein recognition) have become more popular as well. Since the first commercial contactless palm vein acquisition device from Fujitsu [1] became available in 2003, vascular pattern based biometrics have been employed in several application areas, especially in the banking area [2,3]. Vascular pattern based biometrics have several advantages over, for example, fingerprints [4]. This biometric trait is based on the patterns formed by the blood vessels, located underneath the skin, that is, it is an internal biometric trait. While fingerprints are susceptible to dirt and moisture on the skin, skin damage and abrasion, the vascular patterns are assumed to be insensitive to these skin conditions. Furthermore, vascular pattern based biometrics are more resistant to presentation attacks and forgery than are fingerprints and face [4] as the blood vessels are located beneath the skin and are only visible in near-infrared light.

### 1.1. Acquisition Principle and Capturing Devices

To render the patterns formed by the blood vessels visible, special acquisition devices are necessary. These devices are usually denoted as biometric scanners or sensors. The haemoglobin contained in the blood, which is flowing through the veins and arteries, has a higher light absorption coefficient in the near-infrared (NIR) wavelength spectrum (between 700 and 950 nm) than the surrounding tissue. Hence, the vascular pattern can be rendered visible by applying an NIR light source and capturing images using an NIR sensitive camera, which resembles the main parts of a finger or hand vein scanner. There are two distinct configurations depending on the relative positioning of the light source and the camera—light transmission and reflected light (see Figure 1 for an illustration). In the reflected light set-up, the camera and the light source are positioned on the same side of the finger/hand, whereas in the light transmission set-up, both are positioned on opposite sides of the finger/hand. A further distinction can be made regarding the side of the finger/hand which is captured—palmar or dorsal. Palmar refers to the bottom side of the finger/hand, while dorsal images are captured from the top side.

In both finger and hand vein recognition, usually the palmar side is utilised. While reflected light is the preferred set-up in hand vein recognition, finger vein scanners mainly capture the images using light transmission or light dispersion [5,6]. These days there are several commercial off-the-shelf (COTS) solutions for hand as well as finger vein recognition available for a wide range of application scenarios, from securing a personal computer over additional authentication at an automated teller machine (ATM) to high security access control systems at industry buildings. However, most of the COTS solutions have one major drawback for academia and research—the COTS scanners do not output the raw vein images. Instead, they only provide a template, encoded in a proprietary format which is defined by the manufacturer. These templates can only be used with the software provided by the manufacturer, hence limiting the use of those devices in research. Thus, research institutions began to construct their own, custom capturing devices for finger and hand vein images.

The main contribution of this work is the design of such a capturing device. We propose a fully contactless, combined finger and hand vein capturing device and the publication of a vascular pattern dataset, acquired with this device. Contactless acquisition devices have several advantages over touch based ones. The main advantage is that contactless devices achieve a higher user acceptability, mainly due to hygienic reasons and easier handling of the devices. Moreover, contactless acquisition preserves the vascular patterns from distortions [4]. On the other hand, contactless acquisition introduces some challenges as well—due to the higher degree of freedom in terms of finger/hand movement, the physical device design as well as the processing of the vascular pattern images has to account for different types of distortions/artefacts resulting from the image acquisition, including longitudinal finger rotation [7], finger bending and tilts as well as all kinds of translations and rotations of the fingers/hand. Besides these types of misplacements, one of the main challenges is to provide a uniform illumination within the whole range of the allowed relative position of the finger/hand to the capturing device. In the following we give an overview of related work on finger as well as hand vein capturing devices.

### 1.2. Related Work

As the proposed capturing device design is a contactless one, we focus on contactless finger and hand vein capturing devices. While contactless acquisition has become common practice in hand vein recognition, the majority of the capturing devices in finger vein recognition are still contact based.

Almost all of the widely employed COTS finger vein capturing devices are contact based ones, capturing the finger from the palmar view using light transmission or light dispersion. The two major companies providing finger vein authentication solutions are Hitachi Ltd. (Tokyo, Japan) and Mofiria Ltd (Tokio, Japan). The most commonly used devices include the Hitachi H-1 USB finger vein scanner [8] and the Mofiria FVA-U3SX [5] as well as the Mofiria FVA-U4ST [6]. As those are commercial products, not many details about their design have been disclosed, except the recognition performance according to the manufacturers’ data sheets. Due to the challenges and problems with contactless acquisition, there are only a few contactless finger vein capturing devices proposed in research. One of these devices is a mobile finger vein scanner for Android proposed by Sierro et al. [9]. Their prototype device captures contactless palmar finger vein images using reflected light. The illumination source consists of 12,850 nm LEDs, organised in 3 groups of 4 LEDs each (wide angle VSMG3700 and SFH4059 LEDs), providing global as well as optimised homogeneous illumination compensation. The power of each LED group can be adjusted using the Android software. The camera is a low-cost OV7670 colour one, using a CMOS sensor and a wide angle 2.1 mm lens with a maximum resolution of 640×480 pixel. They used an additional NIR pass-through filter with a cut-off wavelength of 740 nm. Another contactless finger vein capturing device was proposed by Kim et al. [10]. This device is based on NIR lasers and uses light transmission. The NIR lasers are manufactured by Lasiris Laser in StokerYale, Canada. A laser line generator lens (E43-475 from Edmund optics in Singapore) with a fixed pan angle is added in order to generate a line laser from the spot laser and should enable a uniform illumination along the finger. The image sensor is a GF 038B NIR CCD (charge coupled device) Camera from ALLIED Vision Technologies, Germany, which is equipped with an additional IR-pass filter. No further details about this device are available, the authors do not even include an image showing their capturing device in the paper. Another contactless device is proposed by Raghavendra et al. [11]. Their low-cost capturing device is able to acquire palmar finger vein images using light transmission as well as fingerprint images in a contactless manner and consists of a NIR light source, a physical structure to achieve a sufficient light intensity, a visible light source and a camera including a lens. The NIR light source is composed of 40 TSDD5210 NIR LEDs with a peak wavelength of 870 nm. The physical structure to achieve a sufficient illumination is wrapped with aluminium foil. The camera is a DMK 22BUC03 monochrome CMOS camera equipped with a T3Z0312CS 8 mm lens. The maximum resolution is 744×480 pixel. Even though the device is a contactless one, the images of the capturing device in the paper reveal that the range of motion for the finger is quite limited in every direction (x, y and z) due to the small opening of the device where the finger has to be placed in. None of the above mentioned capturing devices uses a special NIR enhanced camera. Thus, the resulting image quality in combination with an NIR light source is limited. A more recent device was proposed by Matsuda et al. [12]. It is a contactless walk-through style device which allows to capture multiple fingers at once in real time. It consists of an NIR camera and a depth camera, arranged below the finger placing part and an adaptive, multi-light source arranged vertically on the side of the finger placing part. No further technical details about this device are available but there is an official website from Hitachi [13] showing some images of the sensor prototype.

In the early stage of hand vein recognition, most capturing devices used almost closed box devices having a glass plate and some kind of pegs to force the hand to be placed in a defined position [14,15]. The users found this way of providing their biometric inconvenient and thus, the capturing devices developed from semi contactless ones (e.g., only using some hand attachment or guide [16,17] or a glass plate only [18]) to fully contactless ones. The following review of contactless hand vein capturing devices is not exhaustive but shall provide an overview over the major types of different device designs. The most well-known COTS hand vein authentication system is Fujitsu’s PalmSecure™ [19] one. Their capturing device [20] is contactless and small sized: 35 × 35 × 27 mm. There are many non-commercial devices which have been proposed in several research papers as well, for example, the capturing device originally used to acquire the CASIA Multi-Spectral Palmprint Image Database V1.0 [21]. This device captures palmar hand images using six different wavelengths. It is a box with an opening in the front where the data subject has to put the hand inside. The CCD camera is located at the bottom of the device and the LEDs in different wavelength spectra are located around the camera. Sierro et al. [9] also proposed two contactless palm vein capturing device prototypes. Both are using the reflected light set-up and are equipped with ultrasonic sensors to measure the distance between the camera and the hand. The first prototype uses 20,940 nm LEDs (TSAL6400) as a light source and a Sony ICX618 CCD camera in combination with a 920 nm long-pass filter. The second prototype is able to capture multi-spectral images and uses an additional PTFE (Teflon) sheet to achieve a more uniform illumination. Michael et al. [22] proposed a low-cost contactless capturing device. It has one NIR and one visible light camera to capture both, palm vein and palm print images. The NIR camera has a NIR pass-through filter with a cut-off wavelength of 900 nm. The light source consists of 3 rows of 8 NIR LEDs and 3 yellowish light bulbs to capture the palm prints. The light source is covered by a diffusor paper. Zhang et al. [23] presented an approach to match hand veins using 3D point cloud matching. They use a binocular stereoscopic vision device as contactless capturing device. The hand is place above an NIR light source, consisting of 850 nm LEDs. Dorsal hand vein and knuckle shape images are captured by two NIR sensitive CCD cameras in a stereoscopic set-up, both having an additional NIR pass-through filter. Fletcher et al. [24] developed a mobile hand vein biometric system for health patient identification. They proposed two capturing devices; the first one uses an android smart phone in combination with a rechargeable 850 nm LED light source. The second one employs a low-cost webcam (Gearhead WC1100BLU USB) with integrated 940 nm LEDs and an optical filter, which is powered and controlled by an Android tablet. Both acquire contactless palmar hand vein images. Debiasi et al. [25] presented an illumination add-on for mobile hand vein image acquisition. This device can be used in combination with a modified smart phone (NIR blocking filter removed) to acquire contactless hand vein images from the palmar as well as the dorsal side. They also published a dataset containing palmar and dorsal hand vein images in the scope of the PROTECT Multimodal Biometric Database [26].

While most of the above mentioned capturing device designs are based on low-cost modified cameras, our design is based on a special NIR-enhanced industrial camera in combination with an optimal lens and an additional NIR pass-through filter to reduce image distortions and achieve the best possible image quality. Furthermore, in contrast to other existing designs we employ NIR laser modules instead of LEDs which enable a higher range of finger movement without impacting the image quality. Our capturing device is the first of its kind, able to use light transmission as well as reflected light. Moreover, it is the first combined capturing device, able to acquire finger as well as hand vein images. Finally, we do not only present a new capturing device design including all its technical details, but we also publish a corresponding dataset together with image quality and baseline recognition performance evaluation results on that dataset, which makes this work particularly valuable in the field of finger and hand vein recognition.

### 1.3. Main Contributions

The main contributions of this paper are:Design of a novel fully contactless combined finger and hand vein capturing device featuring laser modules instead of NIR LEDs, a special NIR enhanced industrial camera with an additional NIR pass-through filter to achieve the best possible image quality, an optimal lens and distance between the finger/hand and the camera to allow for minimal image distortions as well as an automated illumination control to provide a uniform illumination throughout the finger/hand surface and to arrive at the best possible contrast and image quality.Publication of all major technical details of the capturing device design—in this work we describe all the major components of the proposed capturing device design. Further technical details are available on request, which makes it easy to reproduce our design.Public finger and hand vein image database established with the proposed capturing device—together with this paper we publish the finger and hand vein datasets acquired with the proposed capturing device. These datasets are publicly available free of charge for research purposes and the finger vein one is the first publicly available contactless finger vein recognition dataset. Due to the nature of contactless acquisition, these datasets are challenging in terms of the different types of the finger/hand misplacements they include.Evaluation of the acquired database in terms of image quality and biometric recognition performance—the images acquired with our sensor are evaluated using several image quality assessment schemes. Furthermore, some well-established vein recognition methods implemented in our already open source vein recognition framework are utilised to evaluate the finger and hand vein datasets. This ensures full reproducibility of our published results. The achieved recognition performance during our evaluation is competitive with other state-of-the-art finger and hand vein acquisition devices, validate the advantages of our proposed capturing device design and prove the good image quality and recognition performance of our capturing device.

The remainder of this paper is organised as follows: Section 2 explains our proposed contactless finger and hand vein capturing device design, introduces the dataset acquired with the help of the proposed capturing device and it explains the experimental set-up, including the utilised recognition tool-chain, the evaluation protocol and the processing of the captured vein images. Section 3 lists the evaluation results of both, the acquired finger and hand vein dataset in terms of image quality and recognition accuracy, as well as the recognition accuracy of the considered fusion combinations. A discussion of the evaluation results, including a comparison with recognition performance results achieved by other capturing devices is provided in Section 4. Section 5 concludes this paper and gives an outlook on future work.

## 2. Materials and Methods

As mentioned in the introduction, a typical finger or hand vein capturing device consists of an NIR sensitive camera and some kind of NIR light source. In the following, the general design of our proposed contactless finger and hand vein capturing device as well as all the individual parts, including technical details and the design decisions are given. Afterwards, the acquired dataset and the utilised biometric recognition tool-chain are described.

### 2.1. Contactless Finger and Hand Vein Acquisition Device

Figure 2 shows our contactless finger and hand vein capturing device with all its individual parts annotated. It consists of an NIR enhanced camera together with a suitable lens and an additional NIR pass-through filter, two NIR illuminators, one laser module based for light transmission as well as one NIR LED based for reflected light, an illumination control board, a touchscreen display to assist the user during the acquisition process and its metal frame together with the wooden housing parts. All its parts are either standard parts which can be easily bought at a local hardware store or custom designed parts which are either laser cut plywood or 3D printed plastic parts and can easily be reproduced as well. The 3D models and technical drawings of those parts are provided on request. The following list summarises the main advantages and differences of our proposed design over the existing ones presented in Section 1.2.

Reflected light as well as light transmission—it is the first acquisition device of its kind, able to acquire reflected light as well as light transmission images. This extends the range of possible uses of this capturing device and speeds up the acquisition process if both types of illumination set-ups shall be investigated.Suitable for finger as well as hand vein images—it is possible to acquire palmar finger as well as hand vein images with the same device. Again, this is the first capturing device able to acquire both using the same device. In the default configuration, finger vein images are captured using light transmission while hand vein images are captured using reflected light but this can be changed in the set-up so there is a high flexibility in terms of possible acquisition configurations.NIR laser modules for light transmission illumination—the application of NIR laser modules has not been that common in finger vein recognition so far. In a contactless acquisition set-up, laser modules exhibit several advantages over LEDs, especially if it comes to increased range of finger/hand movement as well as an optimal illumination and image contrast [27]. Hence, we decided to equip our capturing device with NIR laser modules.Illumination control board and automated brightness control algorithm—the integrated brightness control board handles the illumination intensity of both, the light transmission and the reflected light illuminators. Each of the laser modules in the light transmission illuminator can be brightness controlled separately and independent from the others. This illumination control in combination with our automated brightness control algorithm enables an optimal image contrast without having the operator do any manual settings.Special NIR enhanced industrial camera—our capturing device uses a special NIR enhanced industrial camera. In contrast to modified (NIR blocking filter removed) visible light cameras, those NIR enhanced camera have an increased quantum efficiency in the NIR spectrum. This leads to a higher image contrast and quality compared to cheap, modified visible light cameras.Optimal lens set-up and distance between camera and finger/hand—in contrast to many other, mainly smaller devices (in terms of physical size of the device), we decided to use a lens with a focal length of 9 mm. This allows for minimal image distortions all over the image area, especially at the image borders at the cost of an increased distance between the camera and the finger/hand. Hence, our capturing device is rather big compared to others.Easy to reproduce design—in contrast to most other proposed capturing devices, for which only very few details are available, we provide references to the data sheets and technical details of all of the capturing device’s parts. Furthermore, we provide the 3D models and technical drawings for the frame parts and the 3D printed parts on request. Hence, it is easy to reproduce our proposed capturing device design.Fast data acquisition—due to the automated brightness control and the automated acquisition process, sample data acquisition is fast. Capturing a hand vein image only takes less than a second and capturing a finger vein images takes between 2–4 s once the data subject placed their finger/hand.Ease of use during data acquisition—in contrast to other available vein capturing devices, for our proposed device the data subjects do not need to align their fingers/hands with some contact surface or pegs. This is one of the main advantages of our contactless design, making the data acquisition easier for the data subjects as well as for the operators. The automated illumination control algorithm and the intuitive graphical capturing software further contribute to a smooth and easy data acquisition process. Moreover, the integrated touchscreen display assists the data subjects by indicating which finger/hand to place at the sensor, how to place it and indicates potential misplacements.Biometric fusion can be employed to increase the recognition performance—our proposed capturing device acquires finger vein images as well as hand vein images using two different wavelengths of illumination. Hence, it is easily possible to increase the recognition performance by applying biometric fusion at sensor level with different fusion combinations. An evaluation of selected combinations is done in Section 3.3.

After this general overview of our capturing device we now describe its individual parts.

#### 2.1.1. Camera, Lens and Filter

The camera is an IDS Imaging UI-ML3240-NIR [28] with a maximum resolution of 1280×1024 pixels and a maximum frame rate of 60 fps. It is based on the EV76C661ABT CMOS monochrome image sensor, having a colour depth of 8 bit, a maximum resolution of 1.31 Megapixels, with a pixel size of 5.3 µm and a sensor diagonal of 1/1.84 inches. The main advantage of this camera compared to modified webcams and other visible light cameras is that it is an NIR enhanced industrial camera, which is specifically designed to achieve a high quantum efficiency within the NIR spectrum. Due to its increased NIR sensitivity, an NIR enhanced camera achieves a higher image contrast in the NIR spectrum than a visible wavelength one, which is shown in Figure 3 left, depicting its quantum efficiency chart. The peak wavelengths of our NIR LEDs (850 nm + 950 nm) and NIR laser modules (808 nm) are within the increased sensitivity range of the image sensor.

The camera is equipped with a Fujifilm HF9HA-1B 9 mm fixed focal lens [29]. A lens with an increased focal length has less image distortions but requires a larger distance from the finger, thus increasing the overall size of the scanner. A shorter focal length reduces the minimum distance to the finger but increases the image distortions, especially at the image boundaries. Thus, we decided to use a 9 mm focal length as the best trade-off between the distance to the finger, that is, the overall scanner dimensions and the image distortions introduced due to the lens. A MIDOPT FIL LP780/27 [30] NIR pass-through filter is mounted on top of the lens to further suppress the negative influence of ambient light. The filter transmission chart is depicted in Figure 3 on the right.

#### 2.1.2. Light Sources—Reflected Light and Light Transmission

The capturing device uses two different light sources—a light transmission and a reflected light one. The light transmission illuminator consists of 5 laser diodes [31] including an adjustable constant-current laser diode driver printed circuit board (PCB) [32] and a TO-18 housing with a focus adjustable lens [33] for each of the laser modules (the combination of laser diode + control PCB + housing is denoted as laser module or laser). The laser diodes have a peak wavelength of 808 nm and an optical output power of 300 mW. Each laser module can be brightness controlled separately. The main advantages of the laser modules over LEDs is their higher optical output power and their narrow radiation half angle. This enables a higher degree of vertical finger movement without degrading the image quality [27].

The reflected light illuminator consists of 2 individual illuminators, one at each side of the camera (left and right). Each illuminator is composed of two rows of 8 LEDs each. The first row consists of 850 nm LEDs (Osram SFH 4550 [34] with a radiation half angle of ±3° and a max. radiant intensity of 700 mW/sr). The second row consists of 950 nm LEDs (Vishay Semiconductors CQY 99 [35] with a radiation half angle of ±22° and a maximum radiant intensity of 35 mW/sr). These two types of LEDs have peak wavelengths that are within the recommended spectrum for vascular pattern recognition. Each row can be brightness controlled as well, however only the whole row instead of each individual LED can be set to a certain brightness level. The emission spectra of the 850nm LEDs and the NIR laser modules can be seen in Figure 4, left and right, respectively.

#### 2.1.3. Illumination Control Board and Brightness Control Algorithm

The schematic structure of the control board is depicted in Figure 5. The two main components of the illumination control board are an Arduino Nano board [36] and a Texas Instruments TLC59401 [37]. The Arduino Nano is a complete, breadboard-friendly microcontroller development board based on the Microchip ATmega328P microcontroller [38,39]. The Texas Instruments TLC59401 is an integrated 16-channel LED driver with dot correction and greyscale pulse width modulation (PWM) control enabling a convenient brightness control of LEDs without the need for external components like dropping resistors. Each output can be controlled separately (4096 steps) and has a drive capability of 120 mA. It operates as a constant-current sink and the desired current can be set using one external resistor only. In addition there are external PNP transistors (BC808-25 [40]) to drive the laser modules as their operating current exceeds the maximum current of the TLC59401. The reflected light illuminators are connected to one of the PWM outputs on the Arduino Nano using some external n-channel MOSFET transistors (AO3418 [41]) to drive them. The whole control board is interfaced using a simple, fixed-length, text-based serial protocol to control each of the individual LEDs/laser modules as well as the reflected light illuminators, to set a whole stripe at once and to turn off all illuminators again. On the PC side there is a graphical user interface based capturing control software which facilitates an easy and straight forward data acquisition. At the moment, the capturing process is initiated manually once the data subject placed their hand/finger in the sensor. This process will be automated in the future as well.

The brightness control algorithm controls each of the single light transmission illuminator’s laser modules as well as the reflected light illuminators as a whole. We decided to implement a simple, iterative algorithm based on a comparison against a target grey level, which works as follows—at first the laser centres have to be configured, including the determination of the area of influence for each laser, which is the area in the image a single laser illuminates. Then all lasers are set to an initial intensity level/brightness value which is half of their maximum intensity ($I_{max}$). The live image of the camera is analysed and the current grey level in the circle of influence of each laser is determined ($GL_{current}$) and compared against the set target grey level ($GL_{target}$). The new brightness value is then set according to: $I_{n+1}=I_{n}+I_{corr}$, where $I_{n+1}$ is the new intensity level, $I_{n}$ is the current intensity level and $I_{corr}=\frac{GL_{target}-GL_{current}}{GL_{max}}\cdot \frac{I_{max}}{2\cdot n}$, where $GL_{max}$ is the maximum grey value and *n* is the current iteration. The iteration stops if either the target grey level $GL_{target}$ has been reached or if no more intensity changes are possible. The algorithm finishes in at most $log_{2}(I_{max}$) iterations. Both, the Arduino Nano firmware as well as the capturing software, including our brightness control algorithm are available on request as well.

#### 2.1.4. Frame, Housing and Touchscreen

The outer frame is assembled using Coaxis^®^ [42] aluminium profiles. The Coaxis^®^ system is easy to use with several different profiles and connectors, which can be put together in many different ways. Another advantage is that this system provides a good stability and durability. On top of the aluminium frame there are laser cut plywood (4 mm beech wood) boards as side walls/cover. Figure 2 right shows the outside of the housing including its dimensions. A Waveshare 7inch HDMI LCD (C) touchscreen [43] is located the top front part of the capturing device. This touchscreen is connected to the acquisition PC and displays the live image stream from the camera, including an overlay of the optimal finger/hand position in order to help the data subjects in positioning their fingers/hand and also displays other information about the data acquisition, for example, which finger/hand to place next. The next revision of the capturing device will be a fully embedded one, that is, there is no need for an external PC and the whole data acquisition can be controlled using the device itself with the help of the integrated touchscreen display.

### 2.2. PLUSVein-Contactless Finger and Hand Vein Data Set

To validate our proposed capturing device design and to show the good recognition performance that can be achieved, we established a data set with the help of this device. Due to the contactless acquisition, these datasets are challenging in terms of finger/hand normalisation to compensate for the different types of finger/hand misplacements contained in the data (tilts, bending, in-planar and non-planar rotations). The dataset will be publicly available for research purposes together with the publication of this paper (http://www.wavelab.at/sources/PLUSVein-Contactless/). It contains two subsets—a palmar finger and a palmar hand vein one, including 42 subjects, 6 fingers/2 hands per subject and 5 images per finger/hand in one session. Hence, the finger vein subset contains 1260 images and the hand vein one contains 840 images (2 illumination configurations, 850 and 950 nm, 420 images each) in total. The raw images have a resolution of 1280×1024 pixels and are stored in 8 bit greyscale png format. The visible area of the finger in the images is 600×180 pixels and for the hand 750×750 pixels on average. Some example images are shown in Figure 6. The age and information about the handedness of the data subjects was recorded as well. Besides this information, no other sensitive private information about the subjects was collected. All subjects gave their informed consent for inclusion before they participated in the study. The study was conducted in accordance with the Declaration of Helsinki, and the protocol was approved by the Ethics Committee of the University of Salzburg (PLUSVein Contactless Data Acquisition 2019).

### 2.3. Finger and Hand Vein Recognition Tool-Chain

The recognition tool-chain includes all steps of a biometric recognition system starting with the extraction of the region of interest (ROI) to pre-processing, feature extraction and comparison, which are depicted in Figure 7 and described in the following. In addition, the utilised image quality assessment methods and biometric fusion, especially score level fusion, are explained as well. All of the utilised methods are implemented within our open source vein recognition framework PLUS OpenVein Toolkit (http://www.wavelab.at/sources/OpenVein-Toolkit/).

ROI Extraction

The key aim of the region of interest (ROI) extraction is to select the best suitable image part for the subsequent feature extraction and to automatically normalise the used finger/hand region in a way to avoid shifts, rotations and to account for scale changes. The ROI extraction and finger/hand normalisation is a crucial step, especially in contactless acquisition, to account for the higher degree of freedom and to compensate the different types of finger/hand misplacements. Different ROI extraction methods have been utilised for finger and hand vein images.

For the finger vein images, the finger is aligned and normalised according to a modified version of the method proposed by Lu et al. [44]. This alignment places the finger in the same position in every image, having the same finger width (different scales due to different finger positions). At first the finger outlines (edge between finger and the background of the image) are detected and the centre line (in the middle of the two finger lines) is determined. Afterwards, the centre line of the finger is rotated and translated in a way that it is placed in the middle of the image and the image region outside of the finger is masked out by setting the pixels to black. Then the finger outline is normalised to a pre-defined width. The final step is the actual extraction of a rectangular ROI of a fixed size (450×150 pixels) with its top border located at the fingertip. These steps are visualised in Figure 8.

The ROI method for hand vein images is a modified and extended version of the approach proposed by Zhou and Kumar [45]. At first the hand region is segmented by binarising the image using a local adaptive thresholding technique. Then the local minima and maxima points in the image are found. The local maxima correspond to the finger tips while the local minima correspond to the finger valleys. For the palmar view and the left hand, the second and fourth minima corresponds to the valley between the index and middle finger and the ring and the pinky finger, respectively. A line is fitted between those two valley points and then the image is rotated such that this line becomes horizontal. Afterwards, a square ROI is fitted inside the hand area, with its centre at the centre of mass of the hand (foreground in the segmented image). The size of the square ROI is adjusted such that its size is the maximum square without including any background pixels. The hand ROI extraction steps are shown in Figure 9. As a last step, the ROI image is scaled to a size of 384×384 pixels.

Pre-Processing

Pre-processing approaches try to enhance the low contrast and improve the image quality. Simple **Contrast Limited Adaptive Histogram Equalisation** (**CLAHE**) [46] or other local histogram equalisation techniques are most prevalent for this purpose. Global contrast equalisation techniques tend to over-amplify bright areas in the image while some other dark areas are not sufficiently enhanced. A localised contrast enhancement technique like CLAHE is a suitable baseline tool to enhance the vein images as they exhibit unevenly distributed contrast. CLAHE has an integrated contrast limitation (clip limit) which should avoid the amplification of noise.

**High Frequency Emphasis Filtering** (**HFEF**) [47] tries to enhance the vein images in the frequency domain. At first the discrete Fourier transform of the image is computed, followed by the application of a Butterworth high-pass filter in the frequency domain. Afterwards the inverse Furier transform is computed to give prominence to the vein texture. In order to improve the image contrast the authors also apply a global histogram equalisation as a final step. We applied CLAHE instead of the global histogram equalisation.

**Circular Gabor Filter** (**CGF**) as proposed by Zhang and Yang [48] is another finger vein image enhancement technique which is rotation invariant and achieves an optimal joint localisation in both the spatial and the frequency domain. The authors originally suggested using grey level grouping for contrast enhancement and to reduce illumination fluctuations. Afterwards an even symmetric circular Gabor filter is applied to further attenuate the vein ridges in the image. Gabor filters are widely used to enhance images containing a high amount of texture and to analyse image texture information. In contrast to usual Gabor filters, a CGF does not have a direction, thus it amplifies the vein ridges in each direction. The bandwidth and the sigma of the CGF has to be tuned according to the visible vein information in the images (vein width in pixels).

Furthermore, the images were resized to half of their original size, which not only speeded up the comparison process but also improved the results. For more details on the preprocessing methods the interested reader is referred to the authors’ original publications. Each of the above mentioned pre-processing techniques is at least used for one of the feature extraction methods, but not necessarily with the same parameters for each method. The actual methods and parameters used for each feature extraction method are stated in the settings files (cf. Section 2.4).

Feature Extraction

Three vein pattern based techniques, which aim to extract the vein pattern from the background resulting in a binary image (vein pattern based methods) followed by a comparison of these binary images using a correlation measure and a general purpose key-point based technique were used, which are all algorithms well-established finger vein recognition algorithms.

**Maximum Curvature** (MC [49]) is a curvature based approach which is insensitive to varying vein widths as it aims to emphasise only the centre lines of the veins. At first the centre positions of the veins are extracted by determining the local maximum curvature in cross-sectional profiles obtained by calculating the first and second derivatives in four directions—horizontal, vertical and the two oblique directions. Each profile is classified as either being concave or convex. Vein lines are indicated by local maxima in concave profiles, hence only the concave ones are used. A score is assigned to each centre position which corresponds to the width and curvature of the maxima region. Afterwards, the centre positions of the veins are connected using a filtering operation in all four directions taking the 8-neighbourhood of pixels into account to account for misclassifications at the first step due to noise and other imperfections in the images. The output feature vector is essentially a binary image which is obtained by thresholding the recorded score values using the median of all scores as a threshold.

**Principal Curvature** (PC [50]) is another curvature based approach, which is not based on the derivates but on the gradient field of the image. Hence, the first step is the calculation of the gradient field. Hard thresholding to filter out small gradients by setting their values to zero is performed to prevent amplification of small noise components. Afterwards the normalised gradient field is obtained by normalising the magnitude to 1 at each pixel, which is then smoothed by applying a Gaussian filter. The actual principal curvature calculation is then done based on this smoothed normalised gradient field by computing the Eigenvalues of the Hessian matrix at each pixel. The two Eigenvalues are the principal curvatures and the corresponding Eigenvectors of the Hessian matrix represent the directions of the maximum and minimum curvature. The bigger Eigenvalue corresponds to the maximum curvature among all directions and is recorded and further used. The final step is again a threshold-based binarisation of the principal curvature values to obtain the output feature vector which is essentially a binary vein image.

**Gabor Filter** (GF [4]) is a Gaussian kernel function modulated by a sinusoidal plane wave. Gabor filters are inspired by the human visual system’s multichannel processing of visual information. Several 2D even symmetric Gabor filters with different orientations (in $\frac{\pi }{k}$ steps where *k* is the number of orientations) form a filter bank. The image is filtered using this filter band to extract *k* different feature vectors. The single feature vectors from the previous step are fused and thresholded to get a resulting feature vector. To remove small noise components, this vector is further post-processed using morphological operations, resulting in the final output feature vector, which is again a binary image.

**Scale Invariant Feature Transform** (SIFT [51]) is a key-point based technique. In contrast to the three vein pattern based ones, key-point based techniques use information from the most discriminative points as well as consider the neighbourhood and context information around these points. This is achieved by extracting key-point locations at stable and distinct points in the image and then assigning a descriptor to each detected key-point location. The approach we used is based on the general purpose SIFT descriptor in combination with additional key-point filtering along the finger boundaries. This filtering is done to suppress information originating from the finger shape (outside boundary) instead of the vascular pattern. We originally presented this additional key-point filtering in Reference [52].

Comparison

The three vein pattern based features (MC, PC and GF) are compared using an extended version of the approach proposed by Miura et al. [49]. The input features (binary vein images) are not registered to each and only coarsely aligned (by the preceding ROI extraction). To account for small shifts and rotations, the correlations between the input feature vector and in x- and y-direction shifted as well as rotated versions of the reference feature vector are calculated. The final output score is the maximum among those individual correlation values, representing the best possible overlay/match between the two feature vectors. For the SIFT feature vectors a typical approach for key-point based features is utilised. At first the nearest neighbour for each key-point is found by simply calculating the distance between this key-point and all key-points in the reference feature vector. The nearest neighbours/best correspondences is the one with the highest similarity score. If this score is below a set threshold, the key-point does not have a matching one in the reference feature vector. The final comparison score is the ratio of the matched points and the maximum number of detected key-points in both images (which is the maximum number of possible matches).

Vein Specific Image Quality Assessment

In contrast to fingerprint recognition where there are standardised quality metrics like the NIST Fingerprint Image Quality (NFIQ) [53] and the newer version NFIQ 2.0, there are no standardised metrics in finger- and hand-vein recognition yet. Thus, the finger- and hand-vein images were analysed using GCF [54] as it is a general image contrast metric and hence, independent of the image content. With the help of GCF the image contrast can be quantified exclusively disregarding the actual image content. As we aim to quantify the image quality of vein images, of course two vein specific NIR image quality metrics, namely the approach proposed by Wang et al. [55] (Wang17) and the approach proposed by Ma et al. [56] (HSNR) were included as well. The first approach evaluates the vein image quality fusing a brightness uniformity and a clarity criterion, which is obtained by analysing the local pixel neighbourhoods. The HSNR approach, which is especially tailored for non-contact finger vein recognition, simulates the human visual system by calculating an HSNR index and integrates an effective area index, a finger shifting index and a contrast index to arrive at the final image quality value.

Score Level Fusion

According to the ISO/IEC TR 24722:2015 standard [57], biometric fusion can be regarded as a combination of information from multiple sources, that is, sensors, characteristic types, algorithms, instances or presentations in order to improve the overall system’s performance and to increase the systems robustness. Biometric fusion can be categorised according to the level of fusion and the origin of input data. The different levels of fusion correspond to the components of a biometric recognition system—sensor-level, image-level, feature-level, score-level and decision-level fusion, which are indicated in Figure 7. Sensor-level fusion is also called multisensorial fusion and describes using multiple sensors for capturing samples of one biometric instance [57]. This can either be done by the sensor itself or during the biometric processing chain. Hence, we perform sensor-level fusion as our capturing device acquires finger as well as two different kinds of hand vein images. The actual fusion is done during the biometric processing chain at score level (fusing the output scores of the individual modalities—finger veins, hand veins 850nm and hand veins 950nm).

The following combinations of different acquired modalities are evaluated:Hand veins 850 nm + hand veins 950 nmHand veins 850 nm + finger veinsHand veins 950 nm + finger veinsHand veins 850 nm + hand veins 950 nm + finger veins

Note that, for the combinations including finger veins, only one finger is included in the fusion. We evaluated the combinations including a finger for all fingers of the respective hand and used the best performing finger, which turned out to be the middle finger for both hands. Acquiring images of several, distinct fingers takes more time as only one finger is captured at a time, the same applies to acquiring both hands. Thus, we restricted to the evaluated combinations to the above listed ones which do not considerably increase the acquisition time. The actual score level fusion is performed using the BOSARIS tool-kit [58], which provides a MATLAB based framework for calibrating, fusing and evaluating scores from binary classifiers and has originally been developed for automatic speaker recognition. A 5 fold random split of training and test data with 20 runs was used to train and fuse the scores using BOSARIS. The reported performance results are the average values of the 20 individual runs.

## 2.4. Experimental Setup and Evaluation Protocol

The evaluation is split into three parts—image quality assessment, baseline recognition performance evaluation for the individual subsets and recognition performance evaluation of the fusion combinations. The image quality assessment and the baseline recognition performance evaluation is done separately for the finger dataset and the two hand vein datasets (850 nm and 950 nm illuminator). The three image quality assessment schemes are evaluated for each individual image per dataset. The results are the average values over the whole dataset, that is, there is a single value for the finger vein and the hand vein 850 nm as well as the hand vein 950 nm dataset for each image quality metric. For the recognition performance DET plots as well as the EER (the point where the FMR equals the FNMR), the FMR1000 (the lowest FNMR for FMR = 0.1%) and the ZeroFMR (the lowest FNMR for FMR = 0%) are provided. At first the parameters for the pre-processing and feature extraction are optimised on a training dataset. Each dataset is divided into two roughly equal sized subsets, based on the contained subjects, that is, all fingers/hands of the same person are in one subset. The best parameters are determined on each subset and then applied to the other subset for determining the comparison scores. This ensures a full separation of the training and test set. The final results are based on the combined scores of both test runs. The FVC2004 [59] test protocol is applied for calculating the comparison scores in order to determine the FMR/FNMR: for the genuine scores, all possible genuine comparisons are evaluated, resulting in $n_{gen}=\frac{5\cdot (5-1)}{2}\cdot \left(42\cdot 6\right)=2520$ and $n_{gen}=\frac{5\cdot (5-1)}{2}\cdot \left(42\cdot 2\right)=840$ genuine scores for the finger and hand vein subset, respectively. For the impostor scores only the first template of a finger/hand is compared against the first template of all other fingers/hands, resulting in $n_{imp}=\frac{\left(42\cdot 6)\cdot (42\cdot 6-1\right)}{2}=31,626$ impostor comparisons for the finger vein subset as well as $n_{imp}=\frac{\left(42\cdot 2)\cdot (42\cdot 2-1\right)}{2}=3486$ impostor comparisons for the hand vein ones. The EER/FMR1000/ZeroFMR values are given in percentage terms, for example, 0.47 means 0.47%. The full results including the image quality values for each single image, the comparison scores and plots as well as the settings and script files to reproduce the experiments can be downloaded here: http://www.wavelab.at/sources/Kauba19c/.

## 3. Results

This section presents the results of the image quality assessment as well as the recognition performance evaluation on the acquired datasets and the score level fusion combination of the data sets.

### 3.1. Image Quality Assessment

Table 1 lists the image quality assessment results for the three tested metrics, namely GCF, Wang17 and HSNR. The GCF values range from 0 to 8, the Wang17 values from 0 to 1 and the HSNR values from 0 to 100. Higher values correspond to higher image quality. Note that a cross-modality comparison (finger vs. hand veins) using those metrics does not lead to meaningful results as the underlying input data (images) are fundamentally different. To enable a meaningful quality assessment and a comparison with other, available finger and hand vein dataset, we evaluated several other finger and hand vein datasets by using the same quality metrics. The evaluated finger vein datasets include SDUMLA-HMT [60], HKPU-FID [4], UTFVP [61], MMCBNU_6000 [44], FV-USM [62] and PLUSVein-FV3 [27]. The image quality was evaluated for the following hand vein datasets—Bosphorus Hand Vein [63], Tecnocampus Hand Image [64], Vera Palm Vein [65] and PROTECT HandVein [66]. The discussion of the image quality assessment results is done in Section 4.

### 3.2. Recognition Performance

The recognition performance results should serve as a baseline for further experiments/research conducted on these contactless finger and hand vein datasets. Table 2 lists the performance results in terms of EER, FMR1000 and ZeroFMR where the best results per subset (finger vein, hand vein 850 nm and hand vein 950 nm) are highlighted **bold face**. The corresponding DET plots are shown in Figure 10.

It is evident that MC performed best on all subsets in terms of EER, FMR1000 as well as ZeroFMR except for the finger vein one where it performed second best in terms of EER (but still best in terms of FMR1000 and ZeroFMR). The overall best performance was achieved on the hand vein 850 nm subset using MC and resulting in an EER of 0.35%. In terms of EER, on the finger vein subset SIFT performed best, followed by MC and GF while PC performed worst. On the hand vein 850 nm subset, PC performed second best, followed by SIFT and GF performed worst, while on the 950 nm subset SIFT performed second best, followed by PC and again, GF performed worst.

### 3.3. Biometric Fusion Results

Table 3 shows the results for the tested fusion combination together with the relative performance increase of the combination. The relative performance increase (RPI) refers to the best performing single modality included in the fusion combination (usually the hand vein 850 or hand vein 950 nm result). Each fusion combination improved the results over the respective baseline ones. The overall best results of the tested fusion combinations was the combination of hand vein 850 nm + middle finger achieving an EER of 0.03% which corresponds to a relative performance increase of 1183%. The average improvement in terms of EER (over all feature types) compared to the best baseline (hand veins 850 nm) result for combination 1 is 148%, for combination 2 it is 373%, for combination 3 the average improvement is 140% and for combination 4 it is 365%.

## 4. Discussion

At first, we discuss the image quality of our datasets in comparison with other publicly available finger- and hand-vein datasets. The evaluation results are listed in Table 1. Considering finger veins, our dataset achieved the best results for GCF and HSNR while it is ranked third for Wang17. These results confirm decent image quality in terms of contrast and also a good image quality in terms of vein specific properties. Considering hand veins our 850 nm data set achieved the best results for Wang17 and the second best ones for HSNR, while the 950 nm dataset achieved the best results for HSNR and second best for Wang17. In terms of GCF, both hand vein data sets are only ranked second and third to last, indicating that the general image contrast is lower than for other datasets. However, the vein specific quality metrics still indicate good image quality, despite the inferior image contrast compared to the other datasets. There are several works in the literature [67,68,69,70] that confirm that quality metric results do not necessarily have to correlate with recognition accuracy. The recognition accuracy is the most important aspect of a sensor design and dataset, thus we decided to focus on the recognition accuracy evaluation instead of evaluating the image quality only. Also note that other data set and sensor papers do not report the image quality, thus a direct comparison is not possible based on the image quality.

In the following, we compare our recognition performance evaluation results to other results reported in the literature. Matsuda et al. [12] reported an EER of 0.19% for their walk through style finger vein recognition system. Raghavendra et al. [11] et al. reported an EER of 1.74% for their systems in case of finger veins only. Kim et al. [10] arrived at an EER of 3.6% for their NIR laser based contactless acquisition set-up. Sierro et al. [9] did not present any performance evaluation of their dataset. None of the mentioned authors disclosed their dataset, so their results are not reproducible. As we aim for reproducibility, all the results listed in Table 4 and Table 5 are evaluated on publicly available finger and hand vein datasets and we base our discussion on those results only.

Table 4 lists performance results (in terms of the EER unless indicated otherwise) achieved on various finger vein datasets, ordered by the year of publication, where the last row corresponds to the results presented in this work. The listed results are the best reported ones from the original dataset authors given that they were available and indicated by “-” if they were not available. Note that all of the listed datasets, except the one presented in this work, have been acquired in a non contactless way. Especially compared to the PLUSVein-FV3 dataset, which has been acquired using the same type of NIR enhanced camera and NIR laser modules, the results on our proposed dataset are clearly inferior (5.61% EER vs. 0.06% EER). However, given the increased level of difficulty and challenges of this new, contactless finger vein dataset, the results are still in an acceptable range. The proposed acquisition device design and thus, the acquired dataset, allows for more degrees of freedom in terms of finger/hand movement and thus, more unrestricted finger/hand positioning. This introduces different kinds of finger/hand misplacements, including tilts, bending, in-planar and non-planar rotations. Prommegger et al. showed, that especially those kinds of misplacements cause severe performance degradations for other publicly available datasets [7,71], especially for the SDUMLA-HMT finger vein dataset [60]. Also note that our results should only serve as a baseline and allow room for further improvements. We did not apply any special kind of finger misplacement corrections. If advanced correction schemes are applied, the results can of course be improved.

Regarding contactless hand vein recognition, Michael et al. [22] report an EER of 0.71% using palm veins only. Zhang et al. [23] only evaluate the KC value as measure of the registration between two feature vectors and thus, as an indicator for the recognition performance but they did not evaluate the actual recognition performance. The highest KC value they achieved was 1.1039. Fletcher et al. [24] reported an EER of 6.3% for their fully contactless hand vein based system for health patient identification. Table 5 summarises the achieved recognition performance for several publicly available hand vein recognition datasets, where the last row corresponds to the best results we achieved on our proposed contactless one so far. Note that, except for our proposed dataset and the PROTECT Mobile HandVein [25] dataset, all datasets have been acquired in a non contactless way. In the light of that and taking into account that we used only simple but well-established vein recognition schemes, the achieved recognition performance on our dataset is clearly competitive with other results reported in the literature. It is more than ten times better than the results published for the PROTECT Mobile HandVein [25] dataset and the results published by Fletcher et al. [24], and still twice as good as the results reported by Michael et al. [22], even though the only kind of hand normalisation we applied was the adaptive ROI extraction, correcting different scales, that is, different distances between the camera and the hand. No further tilt or non-planar rotation correction was applied. Again, note that our performance results should only serve as a baseline and can of course be improved.

While the achieved baseline results for the contactless hand vein dataset are quite competitive compared to the contactbased hand vein datasets, the results for the contactless finger vein datasets are clearly inferior to the ones that can be achieved for contactbased finger vein recognition. Contactless finger vein recognition is more challenging than contactless hand vein one for several reason. One reason is that the finger has a much smaller area than the palm of the hand. Thus, small misplacements can lead to a reduced visibility of the vein patterns and a reduced image quality in general, making the recognition more difficult. Moreover, the vascular pattern structure within the finger is more fine-grained than within the palm of the hand. Hence, tilts, rotation and bending of the finger have a more severe effect on the acquired images in terms of the resulting distortions present in the image, again leading to complications during the recognition process. These challenges have to be tackled by suitable normalisation and correction schemes in order to improve the recognition performance for the contactless finger vein data.

The sensor level fusion results clearly indicate that by combining different acquisition modes (finger vein, hand vein 850nm and hand vein 950 nm) the recognition performance can be considerably improved. By combining the hand vein images in the two different wavelengths, the average performance improvement over the best baseline one is 148%. By combining one finger and one hand sample, the best results (MC) are improved by 1183% and 171% over the baseline results for 850 nm and 950 nm hand veins, respectively. By combining all three modes the results were improved as well and are more than 8 times better than the best baseline one (MC). Hence, applying sensor level fusion is an easy way to further enhance the recognition performance of our capturing device.

## 5. Conclusions

We proposed a new capturing device, which is able to acquire finger as well as hand vein images in a fully contactless way. Contactless acquisition has many advantages in terms of hygiene and user acceptance. In addition to the design and technical details of the acquisition device, we also provide a novel, contactless finger and hand vein dataset available for research purposes (can be downloaded here: http://www.wavelab.at/sources/PLUSVein-Contactless/). This dataset is the first available contactless finger vein dataset and one of the first available contactless hand vein datasets. It is a challenging dataset due to the contactless acquisition allowing for more unrestricted finger/hand movement and the resulting finger/hand misplacements. An image quality assessment using three vein tailored metrics has been conducted and confirmed the decent image quality which can be achieved using our proposed capturing device. Moreover, a recognition performance evaluation using several well-established vein recognition schemes has been carried out on this dataset in order to provide baseline results for further research. Those baseline results are competitive for the hand vein data (EER of 0.35%) and within range of other biometric technologies for the finger vein data (EER of 3.66%). Furthermore, biometric sensor level fusion experiments have been conducted to show the additional improvement in the recognition performance which can be achieved by combining finger vein and hand vein data (resulting in an overall best EER of 0.03%).

Our future work includes some improvements on the capturing device itself. The next version of the device should be an embedded device, eliminating the need for an additional PC to control the acquisition process. The capturing device has a built-in touch screen display already which can be used to control it via the graphical user interface. The only thing which is currently missing is the porting of the capturing software to an embedded platform like the Raspberry Pi and the automated start of the capturing process once the data subjects placed their finger/hand. Furthermore, we will extend our contactless finger and hand vein dataset. We are currently acquiring additional subjects and plan to enlarge the dataset to include a total of at least 100 subjects by the end of 2019. Moreover, we aim to do a thorough analysis on which types of finger/hand misplacements are present in the dataset, similar to the work has been done for other finger vein datasets [71]. Based on this analysis we will be able to apply different correction and normalisation schemes in order to improve the recognition performance.

## Figures and Tables

**Figure 1 sensors-19-05014-f001:**
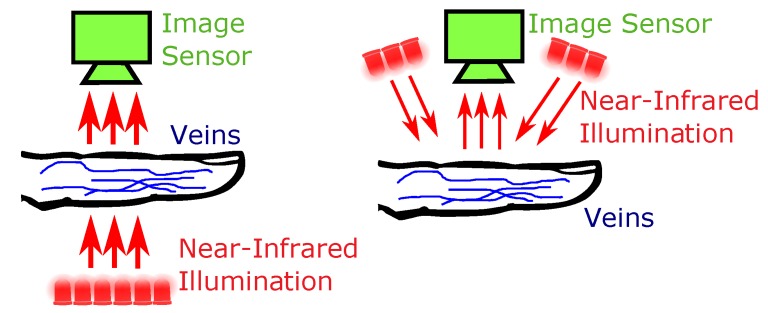
Light source and image sensor positioning, left: light transmission, right: reflected light.

**Figure 2 sensors-19-05014-f002:**
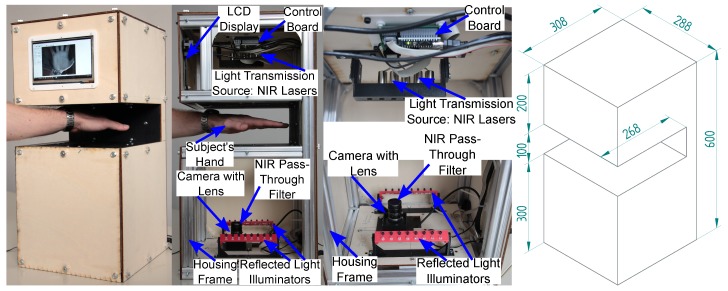
Contactless finger and hand vein capturing device, from left to right: device in use during acquisition, side view with annotated parts, top side view and bottom side detail view, housing including dimensions.

**Figure 3 sensors-19-05014-f003:**
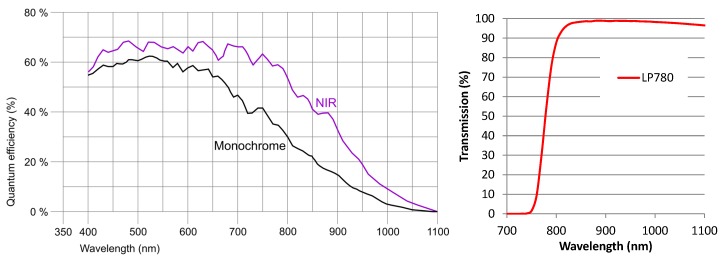
(**Left**) IDS UI-ML-3240-NIR quantum efficiency chart, (**right**) LP780 transmission chart.

**Figure 4 sensors-19-05014-f004:**
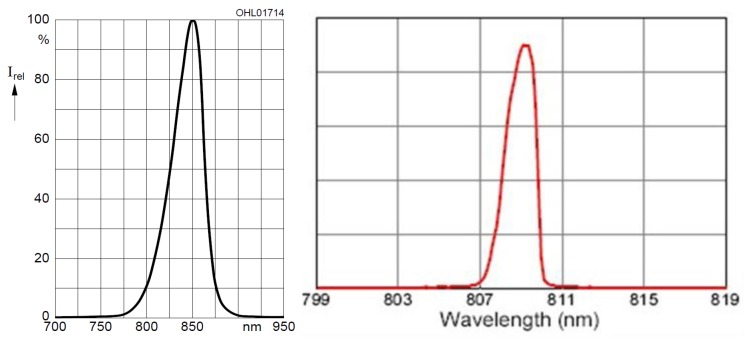
Emission spectrum of the 850nm near infrared (NIR) LEDs (**left**) and the NIR laser modules (**right**), taken from the data sheet [34].

**Figure 5 sensors-19-05014-f005:**
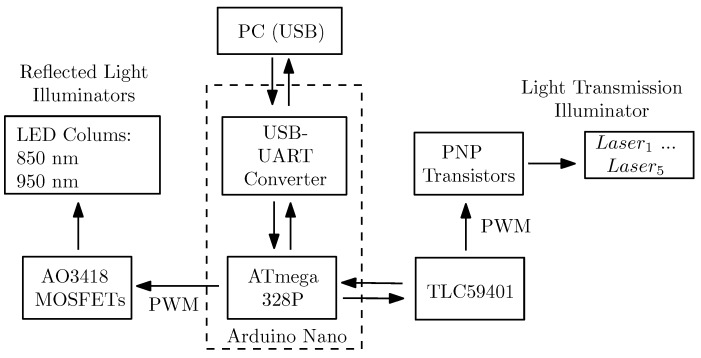
Schematic structure of the illumination control board.

**Figure 6 sensors-19-05014-f006:**
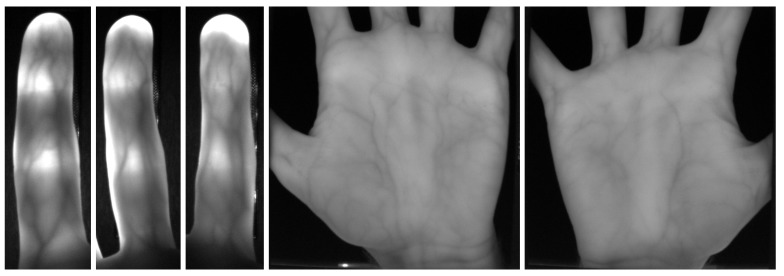
Example images of the PLUSVein-Contactless finger (**left**) and hand (**right**) vein dataset.

**Figure 7 sensors-19-05014-f007:**
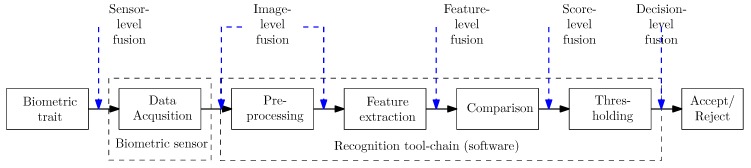
Biometric recognition tool-chain and different levels of biometric fusion.

**Figure 8 sensors-19-05014-f008:**
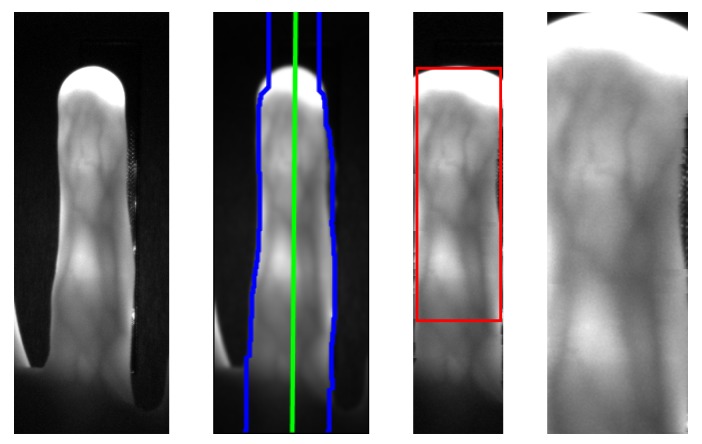
Finger vein region of interest (ROI) extraction process, from left to right: input image, finger outline and centre line detection, finger aligned, masked and normalised ROI boundary, final ROI.

**Figure 9 sensors-19-05014-f009:**
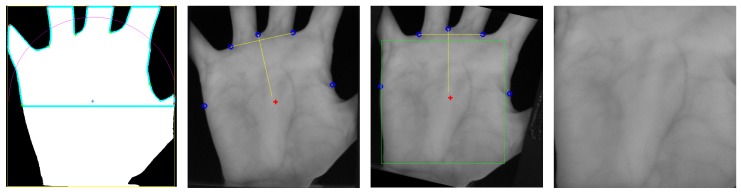
Hand vein ROI extraction process, from left to right: Segmented hand including outline and minima/maxima points, appropriate finger valleys and centre of mass selected, rotationally aligned hand image with maximum possible ROI fitted, final extracted ROI.

**Figure 10 sensors-19-05014-f010:**
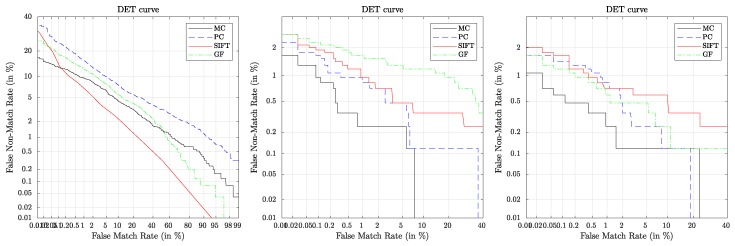
DET plots for finger vein (**left**), hand vein 850 nm (**middle**) and hand vein 950 nm (**right**).

**Table 1 sensors-19-05014-t001:** Image quality assessment results for the proposed datasets (bold face) and several available finger- and hand vein datasets. Best results per quality metric and modality are highlighted **bold face**.

Dataset		GCF	Wang17	HSNR
Finger Vein	**Finger Vein**	**1.72**	0.256	**92.16**
	SDUMLA-HMT [60]	0.986	0.165	80.32
	HKPU-FID [4]	1.46	0.166	88.13
	UTFVP [61]	1.47	**0.356**	87.15
	MMCBNU_6000 [44]	1.52	0.121	87.39
	FV-USM [62]	0.69	0.136	83.35
	PLUSVein-FV3 [27]	1.48	0.306	89.78
Hand Vein	**Hand Vein 850 nm**	1.42	**0.682**	90.43
	**Hand Vein 950 nm**	1.87	0.656	**91.76**
	Bosphorus Hand Vein [63]	2.69	0.329	86.12
	Tecnocampus Hand Image [64]	2.31	0.373	54.33
	Vera Palm Vein [65]	1.31	0.43	85.09
	PROTECT HandVein [66]	**2.8**	0.563	82.43

**Table 2 sensors-19-05014-t002:** Single modality recognition performance results.

Modality		MC	PC	GF	SIFT
Finger Vein	EER [%]	5.61	8.22	6.63	**3.66**
	FMR1000 [%]	**13.12**	23.99	18.39	16.61
	ZeroFMR [%]	**18.75**	42.19	28.76	36.11
Hand Vein 850 nm	EER [%]	**0.35**	0.95	1.55	0.95
	FMR1000 [%]	**0.95**	1.67	2.26	1.9
	ZeroFMR [%]	**1.67**	2.26	2.74	2.74
Hand Vein 950 nm	EER [%]	**0.38**	0.83	0.72	0.82
	FMR1000 [%]	**0.59**	1.43	1.19	1.67
	ZeroFMR [%]	**1.07**	1.67	1.67	2.02

**Table 3 sensors-19-05014-t003:** Score level fusion recognition performance results and improvement over baseline results. Best EER result (combination 2 for MC, combination 1 for PC, 3 for GF and 4 for SIFT) per feature type is highlighted **bold face**.

Combination			MC	RPI	PC	RPI	GF	RPI	SIFT	RPI
1	Hand 850 Hand 950	EER [%]	0.24	44%	**0.16**	405%	0.60	19%	0.37	123%
		FMR1000 [%]	0.36	162%	0.21	586%	0.77	54%	0.65	155%
		ZeroFMR [%]	0.70	139%	4.90	−66%	0.92	82%	1.49	35%
2	Hand 850 Middle Finger	EER [%]	**0.03**	1183%	0.57	65%	0.64	144%	0.48	98%
		FMR1000 [%]	0.01	7862%	0.97	71%	1.19	90%	0.52	268%
		ZeroFMR [%]	0.14	1112%	1.32	72%	1.70	61%	0.79	246%
3	Hand 950 Middle Finger	EER [%]	0.14	171%	0.37	122%	**0.48**	50%	0.26	218%
		FMR1000 [%]	0.14	333%	0.71	102%	0.74	61%	0.37	352%
		ZeroFMR [%]	0.28	289%	1.28	30%	1.11	51%	2.35	−14%
4	Hand 850 Hand 950 Middle Finger	EER [%]	0.04	849%	0.22	272%	0.57	26%	**0.20**	311%
		FMR1000 [%]	0.00	-	0.36	298%	0.62	91%	0.30	449%
		ZeroFMR [%]	0.17	861%	11.47	−85%	0.74	127%	2.08	−3%

**Table 4 sensors-19-05014-t004:** Performance results of other publicly available finger vein datasets, ordered by publication year, ”-” means that this information is not available. The “cla” column indicates contactless acquisition.

Name and Reference	Images/Subjects	cla	Feature Type	Performance (EER)	Year
PKU [72]	50,700/5208	no	WLD [72]	0.87%	2008
THU-FVFDT [73]	6540/610	no	MLD [73]	98.3% ident. rate	2009
SDUMLA-HMT [60]	3816/106	no	Minutia [74]	98.5% recogn. rate	2010
HKPU-FID [4]	6264/156	no	Gabor Filter [4]	0.43% (veins only)	2011
UTFVP [61]	1440/60	no	MC [49]	0.4%	2013
MMCBNU_6000 [44]	6000/100	no	-	-	2013
CFVD [75]	1345/13	-	-	-	2013
FV-USM [62]	5940/123	no	POC and CD [62]	3.05%	2013
VERA FV-Spoof [76,77]	440/110	no	MC [49]	6.2%	2014
PMMDB-FV [26]	240/20	no	MC [49]	9.75%	2017
PLUSVein-FV3 [27]	3600/60	no	MC [49]	0.06%	2018
**Contactless FingerVein**	840/42	**yes**	MC [49]	3.66%	2019

**Table 5 sensors-19-05014-t005:** Related performance results of publicly available hand vein recognition datasets, ordered by publication year. The “cla” column indicates contactless acquisition.

Name and Reference	Images/Subjects	cla	Feature Type	Performance (EER)	Year
CIE [18]	2400/50	no	Thresholding [78]	1.1%	2011
Bosphorus Hand Vein [63]	1575/100	no	Geometry [79]	2.25%	2011
CASIA Multispectral [21]	7200/100	no	LBP/LDP [80]	0.09%	2011
Tecnocampus Hand Image [64]	6000/100	no	BDM [64]	98% recogn. rate	2013
Vera Palm Vein [65]	2200/110	no	LBP [81]	3.75%	2015
PROTECT HandVein [66]	2400/40	no	SIFT [51]	0.093%	2018
PROTECT Mobile HandVein [25]	920/31	**yes**	MC [49]	4.13%	2018
**Contactless HandVein**	420/42	**yes**	MC [49]	0.35%	2019

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
