# Peer review of "Combined Fully Contactless Finger and Hand Vein Capturing Device with a Corresponding Dataset"

_sensors, 2019, doi:10.3390/s19225014_

Round 1

Reviewer 1 Report

In this paper, the authors proposed a device for reflection and transmission imaging of fingers and palms vein in a non-contact manner. The following questions should be answered.

1) It is ambiguous whether this paper proposes a device or a recognition method. Abstract emphasizes new devices and public DB releases, but experimental results focus on recognition performance. If the new device is the main contribution of your paper, the experimental results should be described in terms of the quality of the acquired images, ease of use, and so on. If the recognition accuracy is main contribution, there is nothing new in this paper and no significant influence.

2) Analysis of recognition performance is presented based on single biometric information. However, in this paper, the finger and palm vein images can be taken by reflection and transmission. In order for the proposed method to be consistent with the results, it would be desirable to present multi-modal fusion recognition accuracy through the combination of different modality images.

3) The pre-processing step of vein imaging is no different from the existing methods and is presented too abstractly. Perhaps this is because the new device is the main contribution of your paper.Consequently, this paper adopts the preprocessing and recognition algorithm and performance evaluation protocol proposed in the previous vein recognition study, despite the focus on the new device and database. I believe this should be improved entirely with content that can highlight the merits of the device or newly promoted DB.

Author Response

At first we like to thank the reviewer for their time and valuable comments which helped us to improve the paper. The paper was revised/rewritten and extended with these comments in mind. The changes are highlighted in the new version of the manuscript (added parts are blue and underlined). Please find our detailed response to your comments below.

Point 1: It is ambiguous whether this paper proposes a device or a recognition method. Abstract emphasizes new devices and public DB releases, but experimental results focus on recognition performance. If the new device is the main contribution of your paper, the experimental results should be described in terms of the quality of the acquired images, ease of use, and so on. If the recognition accuracy is main contribution, there is nothing new in this paper and no significant influence.

Response 1: The main contribution of this paper is the capturing device design and the data set acquired with this device. We now added an image quality evaluation of the data set and some comments on the ease of use of our device in comparison to existing finger/hand vein acquisition devices. However, it has to be noted that the image quality does not directly correspond to the recognition performance (see [1] and [2]), but the main performance indicator for a biometric system is its recognition performance. On the other hand, there are no standardised quality metrics for vein based biometrics yet. Moreover, most of the other works about publicly available data sets do report image quality as a performance measure. Hence, a comparison with related work based on the image quality is not feasible. As a consequence, we focused on the evaluation of the recognition performance using already well-established recognition schemes to prove the good image quality and biometric performance that can be achieved using our proposed capturing device. Furthermore, the reported performance results should serve as baseline results for other researchers who are interested in working with this rather challenging data set and aim to improve the recognition performance.

Point 2: Analysis of recognition performance is presented based on single biometric information. However, in this paper, the finger and palm vein images can be taken by reflection and transmission. In order for the proposed method to be consistent with the results, it would be desirable to present multi-modal fusion recognition accuracy through the combination of different modality images.

Response 2: We thank the reviewer for pointing this out. Of course a multi-modality fusion is easily possible. As we have three different modalities, we decided to employ biometric fusion at sensor level. We investigated several different combinations (fusing one finger and one hand modality (for 850 and 950nm), fusing both hand modalities, fusing one finger and both hand modalities) and present the evaluation results in comparison to the baseline, single modality ones. We decided to evaluate those fusion combinations as they do only slightly increase the acquisition time. Fusing left and right hands or fusing several fingers would considerably increase the acquisition time as the data subjects have to reposition their hand/fingers and the capturing process has to be repeated for the additional hand/finger. 

Point 3: The pre-processing step of vein imaging is no different from the existing methods and is presented too abstractly. Perhaps this is because the new device is the main contribution of your paper.

Response 3: As the reviewer already pointed out in his first point, the main contribution of our paper is the new device and the data set. Thus, we only describe the vein processing tool-chain briefly. However, we added some more details about the employed pre-processing methods.

References

[1] HÄMMERLE-UHL, Jutta; POBER, Michael; UHL, Andreas. Systematic evaluation methodology for fingerprint-image quality assessment techniques. In: 2014 37th International Convention on Information and Communication Technology, Electronics and Microelectronics (MIPRO). IEEE, 2014. S. 1315-1319.

[2] HÄMMERLE-UHL, Jutta; POBER, Michael; UHL, Andreas. General purpose bivariate quality-metrics for fingerprint-image assessment revisited. In: 2014 IEEE International Conference on Image Processing (ICIP). IEEE, 2014. S. 4957-4961.

Reviewer 2 Report

The authors have put significant efforts in buiding a biometrics data acquisition device, specifically vein data capturing from finger and hand. 

The device performs very well and the collected data are useful indeed. 

However,

1. only little work is dedicated to the system hardware, in which many readers would be interested. There is lack of information about the hardware of such devices in the literature. 

2. there are some references (2 are given below) that might seem of use for the authors. Some details are given there about the vein image processing methods and also about a simple laboratory approach to collect finger vein data. In the book (Biometrics and Kansei Engineering) some original information was collected from known companies, like Hitachi, and given there.

- Finger Vein Pattern Extraction Algorithm. Proceedings HAIS 2011 – International Conference on Hybrid Intelligent Systems, Wroclaw, May 23-25. LNCS 6678, Springer-Verlag Heidelberg, part I, Germany, 2011, 404–411

- Human Identification by Vascular Patterns. In: Biometrics and Kansei Engineering. Springer Science and Business Media, NY, (2012).

3. I do not know why the authors use 'contact-less' instead of 'contactless'. It is given so even in the title. 

Author Response

We thank the reviewer for their time and valuable comments which helped us to improve the paper. The paper was revised/rewritten and extended with these comments in mind. The changes are highlighted in the new version of the manuscript (added parts are blue and underlined). Please find our detailed response to your comments below.

Point 1: only little work is dedicated to the system hardware, in which many readers would be interested. There is lack of information about the hardware of such devices in the literature. 

Response 1: We are aware that in most other works in the literature the capturing hardware is only mentioned briefly. Thus, we decided to provide all important technical details in our work. We now added some more detailed information about the capturing device hardware itself, in particular about the housing, the principle design of the control PCB and the imaging hardware.

Point 2: there are some references (2 are given below) that might seem of use for the authors. Some details are given there about the vein image processing methods and also about a simple laboratory approach to collect finger vein data. In the book (Biometrics and Kansei Engineering) some original information was collected from known companies, like Hitachi, and given there.

- Finger Vein Pattern Extraction Algorithm. Proceedings HAIS 2011 – International Conference on Hybrid Intelligent Systems, Wroclaw, May 23-25. LNCS 6678, Springer-Verlag Heidelberg, part I, Germany, 2011, 404–411

- Human Identification by Vascular Patterns. In: Biometrics and Kansei Engineering. Springer Science and Business Media, NY, (2012).

Response 2: We thank the reviewer for referring us to those publications. The first one only presents a laboratory designed capturing device and does not give any details about commercial finger vein sensors. There is no image of the device given and no clue that it is a contactless device, hence we did not include it in the related work section. The chapter “Human Identification by Vascular Patterns” in the book “Biometrics and Kansei Engineering” also presents a laboratory prototype finger vein capturing device and some details about the Fujitsu PalmSecure scanner. However, no further technical details about the commercial scanners except their basic working principle, which we already included in our work, are presented. The proposed laboratory device is again no contactless one, thus we did not include this information in our paper as well.

Point 3: I do not know why the authors use 'contact-less' instead of 'contactless'. It is given so even in the title.

Response 3: The auto correction of our tex software suggested to use “contact-less” instead of “contactless”. We now changed it to “contactless” to be consistent with the title.

Round 2

Reviewer 1 Report

In the revised verstion, all of the raised issues were addressed and adequately responsed.